# Design, Synthesis, and Anti-Proliferative Action of Purine/Pteridine-Based Derivatives as Dual Inhibitors of EGFR and BRAF^V600E^

**DOI:** 10.3390/ph16050716

**Published:** 2023-05-08

**Authors:** Samar A. El-Kalyoubi, Hesham A. M. Gomaa, Elshimaa M. N. Abdelhafez, Mohamed Ramadan, Fatimah Agili, Bahaa G. M. Youssif

**Affiliations:** 1Department of Pharmaceutical Organic Chemistry, Faculty of Pharmacy, Port Said University, Port Said 42511, Egypt; 2Department of Pharmacology, College of Pharmacy, Jouf University, Sakaka 72341, Aljouf, Saudi Arabia; 3Medicinal Chemistry Department, Faculty of Pharmacy, Minia University, Minia 61519, Egypt; 4Pharmaceutical Organic Chemistry Department, Faculty of Pharmacy, Al-Azhar University, Assiut Branch, Assiut 11651, Egypt; 5Chemistry Department, Faculty of Science (Female Section), Jazan University, Jazan 82621, Jazan, Saudi Arabia; 6Pharmaceutical Organic Chemistry Department, Faculty of Pharmacy, Assiut University, Assiut 71526, Egypt

**Keywords:** cancer, EGFR, BRAF, anti-proliferative, purine, pteridine, docking

## Abstract

The investigation of novel EGFR and BRAF^V600E^ dual inhibitors is intended to serve as targeted cancer treatment. Two sets of purine/pteridine-based derivatives were designed and synthesized as EGFR/BRAF^V600E^ dual inhibitors. The majority of the compounds exhibited promising antiproliferative activity on the cancer cell lines tested. Compounds **5a**, **5e**, and **7e** of purine-based and pteridine-based scaffolds were identified as the most potent hits in anti-proliferative screening, with GI_50_ values of 38 nM, 46 nM, and 44 nM, respectively. Compounds **5a**, **5e**, and **7e** demonstrated promising EGFR inhibitory activity, with IC_50_ values of 87 nM, 98 nM, and 92 nM, respectively, when compared to erlotinib’s IC_50_ value of 80 nM. According to the results of the BRAF^V600E^ inhibitory assay, BRAF^V600E^ may not be a viable target for this class of organic compounds. Finally, molecular docking studies were carried out at the EGFR and BRAF^V600E^ active sites to suggest possible binding modes.

## 1. Introduction

Enhanced understanding of therapeutic targets plays a significant role in the advancement of new drugs in cancer research. This approach is based on the assumption that altering a particular cancer biomarker will lead to a positive treatment result [1]. The selectivity of anti-cancer drugs can significantly enhance their effectiveness in damaging cancer cells while minimizing adverse reactions on healthy cells [2]. However, due to drug resistance, suppressing just one target often has only a temporary impact. To obtain optimal outcomes, it is essential to target multiple targets simultaneously due to the diversity in cancers [3,4].

One strategy for simultaneously blocking two or more targets is combined chemotherapy. Yet there are frequently discrepancies between the pharmacokinetic properties and metabolic stabilities of two or more medications. Moreover, the use of multiple medications at the same time may result in hazardous medication interactions [5]. These issues might be addressed by combining two drugs within a single molecule that affects multiple targets [6]. Multi-target medications, commonly referred to as “hybrid” molecules, were shaped by fusing two or more distinct pharmacophore moieties into a single molecule. These drugs have attracted a lot of attention lately [7].

It has been demonstrated that kinases control a wide range of essential tumour behaviours, including tumour development, metastasis, neovascularization, and chemotherapeutic resistance. As a consequence, the FDA has recently validated many kinase blockers for use in a wide variety of cancers, making them a key focus of therapeutic development [8].

The established BRAF^V600E^ mutation was expected to be a resistance mechanism after EGFR blocker therapy [9]. The feedback activation of EGFR signalling has also been connected to the resistance that develops in colorectal cancer [10]. Furthermore, EGFR may be activated by BRAF suppression, leading to continuing tumour development [11]. To address these problems, a BRAF/EGFR combination was utilized. Much research in cases of metastatic colorectal cancer with BRAF^V600E^ mutations discovered that the BRAF–EGFR combination might lead to critical therapeutic action [9]. Hence, sequential suppression of the two kinases may provide a solution for the EGFR activation issue.

Several studies have been conducted to investigate the potential activity of 1,3-dimethyl-1*H*-purine-2,6(3*H*,7*H*)-dione derivatives (methylxanthines) on tumour cell molecular aspects and growth [12,13,14]. Theophylline and caffeine, two well-known methylxanthine derivatives, have the ability to suppress cell proliferation in addition to the metastatic behaviour of melanoma cancer cells [15]. Substitution at N-7 and/or C-8 of the xanthine ring subsequently drew the attention of many researchers seeking novel anti-tumour agents [16,17,18,19].

Earlier, we mentioned the synthesis of a series of purine-2,6-dione derivatives with possible anti-proliferative properties, with compound **I** (Figure 1) being the most effective derivative against the investigated cell lines. Compound **I** demonstrated promising EGFR suppressive effect, with an IC_50_ of 0.32 µM [16]. In another series [20], Compound **II** (Figure 1) demonstrated promising anti-proliferative activity with a GI_50_ value of 1.60 µM against four cancer cell lines tested. Compound **II** was tested for EGFR inhibitory activity. The study findings revealed that **II** had an IC_50_ against the target enzyme of 0.30 µM, that is more potent than the reference staurosporine (IC_50_ = 0.4 µM). In contrast to the reference drug methotrexate, compound **III** (Figure 1) demonstrated good anti-proliferative action versus the lung carcinoma cell line (A549), with an IC_50_ value of 12.2 µM [21].

Spiro scaffolds, on the other hand, are an additional class of building block with potential medicinal chemistry features.

During the process of developing a new drug, chemists include a rigid ring to significantly minimize the entropic cost upon binding to the target protein. A spiro ring fusion is another appealing method for achieving conformational restriction.

Their inherent three-dimensional and atypical structural properties make them particularly useful in the exploration and design of novel drugs. Inhibitors of protein–protein interactions (e.g., p53–MDM2 interaction) and enzyme inhibitors (aspartyl proteases, kinases, renin, and BACE1) have effectively included spiro ring structures in recent years [22].

Pteridines are compounds with pyrimido[4,5-*b*]pyrazine rings **IV** (Figure 2). Many living organisms produce these bicyclic compounds, which serve many biological functions. The majority of naturally occurring pteridines are known as pterins **V** (Figure 2) because they have a carbonyl and an amino group at ring positions 4 and 2, respectively (Figure 2) [23].

Due to their significance in both health and sickness, pteridines have long been the subject of medicinal and biomedical chemistry research. In order to target a broad range of human pathologies, such as neoplasms, microbial infections, chronic inflammatory disorders, and others, many pteridine derivatives have been synthesised and evaluated for biological actions. It has shown that these compounds have a high potential for drug development [24]. Zhou et al., prepared and optimised a series of pteridine-7(8*H*)-dione derivatives and assessed their suppressor potential against wild-type epidermal growth factor receptor (EGFR^WT^) and the mutant-type (EGFR^L858R/T790M^). Compound **VI** (Figure 2) was the most effective in the series, suppressing both mutant enzyme EGFR^L858R/T790M^ (IC_50_ = 0.68 nM) and wild EGFR^WT^ (IC_50_ = 1.21 nM) [25].

In the context of cancer, molecular docking can be used to identify potential drugs that could target specific proteins involved in cancer cell growth and survival, such as oncogenes or proteins involved in angiogenesis. By identifying molecules that bind to these targets with high affinity, researchers can develop drugs that specifically target cancer cells, while minimizing side effects. However, it is important to note that molecular docking is only a computational prediction and must be validated experimentally. It is also limited by the accuracy of the protein structure and the quality of the small molecule library used for screening. Therefore, molecular docking should be considered a complementary tool in drug discovery rather than a replacement for experimental methods.

In keeping with our previous investigations on the anti-proliferative aspects of purine-based derivatives [16,20,21], and inspired by the promising anti-proliferative and EGFR inhibitory activities of pteridine derivatives [25], we present the synthesis and design of two series of new compounds, **5a**–**e** and **7a**–**f** (Figure 3). The newly synthesized elements are from two different scaffolds. Scaffold A elements **5a**–**e** were purine-based derivatives with a spiro moiety in their backbone structure. The second series consists of pteridine-based derivatives **7a**–**f**. Four distinct cancer cell lines were used to assess the newly created chemicals’ anti-proliferative ability. Furthermore, the most potent elements from the two series were studied further for their suppressive impact on BRAF^V600E^ and EGFR. Molecular docking analysis was utilised to evaluate how these molecules attach to the active sites of BRAF^V600E^ and EGFR.

## 2. Results and Discussion

### 2.1. Chemistry

Figure 1 depicts the synthetic route used to synthesize purine-based derivatives **5a**–**e**. In situ, nitrosation of 6-amino-1-alkyluracils **1a**–**f** [21,26,27] with HNO_2_ afforded compounds **2a**–**f** in high yields, which were then reduced with ammonium sulphide to produce 5,6-diaminouracils **3a**–**f**. The nucleophilic attack of the amino group of diaminouracils **3a**–**e** on the carbonyl group of 2,7-dibromo-9*H*-fluoren-9-one (**4**) takes place to form intermediate **VII** followed by the elimination of a water molecule to form intermediate **VIII** (Figure 2), which underwent intramolecular aza-Michael addition that resulted in the formation of compounds **5a**–**e** in reasonable yields (59–68%). The structures of compounds **5a**–**e** were completely consistent with their ^1^H NMR, ^13^C NMR, mass spectra, and elemental analyses, with compound **5a** used as an example to discuss structure confirmation. The ^1^H NMR spectrum of **5a** revealed the disappearance of the signals of the 5,6-diamino-groups at δ*_H_* 6.0–7.5 ppm and the appearance of characteristic protons of the two NH of the dihydropurine ring at δ*_H_* 7.50 and 7.19 ppm. In addition to the appearance of a spiro carbon characteristic signal in the ^13^C NMR spectrum at δ 102.1 ppm. Compound **5a** has a molecular weight of 476 based on elemental analysis. The molecular ion peak in the mass spectrum of **5a** corresponds to the molecular weight of *m*/*z* = 476, with the appearance of M^+^ + 2 at *m*/*z* = 478 and M^+^ + 4 at *m*/*z* = 480.

Figure 3 describes the synthesis of acenaphtho [1,2-*g*] pteridines **7a**–**f**. Ram and Pandy previously prepared compounds **7a** and **7d** by dissolving diaminouracil hydrochloride salts in water and then refluxing with acenaphthoquinone **6** in acetic acid for 6 h [28]. In the current study, we prepared compounds **7a**–**f** by condensation of 5,6-diaminouracils **3a**–**f** with acenaphthoquinone (**6**) under reflux conditions for 4 h in the presence of catalytic amounts of acetic acid (yields 60–71%). Another method for preparing compounds **7a**–**f** was to heat under fusion 5,6-diaminouracils **3a**–**f** with acenaphthoquinone (**6**) for 15 min in presence of drops of DMF, which resulted in slightly higher yields (69–79%). Compounds **7a**–**f** elemental analyses, NMR, and mass spectra all agreed with the assigned product structures. The disappearance of the two NH_2_ group signals of uracils and the appearance of the deshielded aromatic protons was revealed by ^1^H NMR. Furthermore, the characteristic signal of uracil NH was found at δ 11.63–12.16 ppm, as well as thiouracil NH at δ 13.09 ppm.

### 2.2. Biology

#### 2.2.1. Cell Viability Assay

To evaluate the survivability of novel substances, the human mammary gland epithelial (MCF-10A) cell line was utilized [29,30]. The vitality of compounds **5a**–**e** and **7a**–**f** was assessed using the MTT method after incubation on MCF-10A cells for four days. Cell viability at 50 µM was more than 88% for all of the agents evaluated, according to Table 1, and none of the substances evaluated had any harmful impacts.

#### 2.2.2. Anti-Proliferative Assay

With erlotinib serving as the reference medication, the MTT assessment was utilized to evaluate the anti-proliferative effect of **5a**–**e** and **7a**–**f** against four human cancer cell lines: HT-29 (colon cancer cell line), Panc-1 (pancreatic cancer cell line), A-549 (lung cancer cell line), and MCF-7 (breast cancer cell line) [31,32,33]. Table 1 reveals the median inhibitory concentration (IC_50_).

Compounds **5a**–**e** “Scaffold A” and **7a**–**f** “Scaffold B” demonstrated promising antiproliferative effect, with GI_50_ ranging from 38 nM to 101 nM for **5a**–**e** and 44 nM to 92 nM for **7a**–**f**. All were less effective than the reference erlotinib (GI_50_ = 33 nM). With a GI_50_ value of 38 nM against the four cancer cell lines evaluated, molecule **5a** (R = Me, X = O, Scaffold A) was the most effective of all synthesized derivatives and was equivalent to the reference drug erlotinib.

Substitution of the oxygen atom at position 2 of compound **5a** with a sulphur atom results in compound **5c** (R = Me, X = S, Scaffold A), which has significantly reduced anti-proliferative activity, with a GI_50_ value of 101 nM, being 2.7 times less effective than **5a**, suggesting the significance of the oxygen atom at position 2 of “Scaffold A” compounds for anti-proliferative action. The same pattern holds true when the methyl group at position 3 of compound **5a** is replaced with an ethyl group as in molecule **5b** (R = Et, X = O, Scaffold A), resulting in a marked reduction in anti-proliferative action with a GI_50_ of 86 nM, making **5b** 2.3-fold less effective than **5a**. These results revealed the significance of both the oxygen atom in the second position and the methyl group in the third position of Scaffold A compounds for antiproliferative activity.

Surprisingly, compounds **5d** (R = Bn, X = O, Scaffold A) and **5e** (R = 2-Cl-Bn, X = O, Scaffold A) where the methyl group in compound **5a** has been replaced by benzyl and 2-chlorobenzyl moieties, respectively, revealed encouraging anti-proliferative action, with GI_50_ values of 50 nM and 46 nM, respectively, being 1.4-fold and 1.2-fold less effective than **5a**, but much stronger than the ethyl derivative, **5b** (GI_50_ = 86 nM). These findings suggest that the nature of the third-position substitution in Scaffold A compounds plays a significant role in anti-proliferative activity, with activity increasing in the order: methyl > 2-chlorobenzyl > benzyl > ethyl.

As previously stated, “Scaffold B” compounds **7a**–**f** demonstrated moderate antiproliferative effect, with GI_50_ values ranging from 44 nM to 92 nM. Compound **7e** (R = 2-chlorobenzyl, X = O, Scaffold B) was the most significant derivative in this series, with a GI_50_ value of 44 nM versus the four cancer cell lines evaluated, but it was 1.3-fold less effective than the reference erlotinib. Molecule **7e** was similar to its congener **5e**, which has the same substitution pattern but with Scaffold A” in its backbone structure. 

Compound **7a** (R = Me, X = O, Scaffold B) demonstrated moderate anti-proliferative action with a GI_50_ value of 58 nM, being 1.5-fold less potent than its congener **5a** (R = Me, X = O, Scaffold A). Once again, the replacement of the methyl group in **7a** with the ethyl group in **7b** (R = Et, X = O, Scaffold B) led to a reduction in anti-proliferative effect, with a GI_50_ value of 67 nM for **7b**. Furthermore, replacing the oxygen atom with a sulphur atom, as in compound **7c** (R = Me, X = S, Scaffold B), reduces activity, with **7c**; GI_50_ equal to 81 nM.

Compounds **7d** (R = benzyl, X = O, Scaffold B) and **7e** (R = 2-chlorobenzyl, X = O, Scaffold B), in which the methyl group in **7a** was replaced by benzyl and *o*-chlorobenzyl moiety, respectively, demonstrated a significant difference in anti-proliferative activity. Compound **7d** demonstrated a marked decrease in anti-proliferative action with a GI_50_ value of 92 nM, being 1.5-fold less effective than **7a**, whereas **7e** outperformed **7a** in activity with a GI_50_ value of 44 nM.

Finally, the unsubstituted derivative, **7f** (R = H, X = O, Scaffold B), showed weak anti-proliferative action with a GI_50_ value of 76 nM, being 1.3-fold less effective than the methyl derivative, **7a** (R = Me, X = O, Scaffold A), indicating that the free NH group in the third position is not favoured for activity. Unfortunately, due to a lack of a sufficient number of compounds, such a rule cannot be generalized, necessitating further research on this topic in the future.

#### 2.2.3. EGFR Inhibitory Assay

The most potential anti-proliferative derivatives, **5a**, **5d**, **5e**, **7a**, and **7e**, were further assessed for their suppressive effect against EGFR, as a possible molecular target for their mechanism of action [34,35]. Table 2 lists the IC_50_ values against erlotinib, which was utilized as a reference.

The compounds assessed revealed promising EGFR inhibitory action, with IC_50_ values ranging from 87 nM to 112 nM, in contrast to erlotinib that has an IC_50_ value of 80 nM. The results of this assay are the same as the results of the anti-proliferative assay, where compound **5a** (R = Me, X = O, Scaffold A), the most potent anti-proliferative agent, was determined to be the most effective EGFR suppressor, with an IC_50_ value of 87 ± 07 nM, equal to erlotinib (IC_50_ = 80 nM).

Molecules **7e** (R = 2-chlorobenzyl, X = O, Scaffold B) and **5e** (R = 2-chlorobenzyl, X = O, Scaffold A) ranked second and third in terms of action with comparable IC_50_ values of 92 ± 07 and 98 ± 08, respectively. Finally, compounds **5d** (R = benzyl, X = O, Scaffold A) and **7a** (R = Me, X = O, Scaffold B) showed weak EGFR suppressive action with IC_50_ values greater than 100 nM. These findings imply that EGFR can be a potential target for compounds **5a**, **5e**, and **7e**, which required more in-depth structural investigation to obtain a lead compound for future development.

#### 2.2.4. BRAF^V600E^ Inhibitory Assay

Molecules **5a**, **5d**, **5e**, **7a**, and **7e** were further explored as potential BRAF^V600E^ inhibitors [36]. Table 2 shows the IC_50_ values in comparison to erlotinib that was utilized as a control. Findings from Table 2 revealed that the assessed molecules had weak BRAF^V600E^ suppressive action, with IC_50_ values ranging from 92 nM to 183 nM, being at least 1.5-fold less effective than erlotinib (IC_50_ = 60 nM). Once again, compound **5a**, the most potent derivative in both the anti-proliferative assay and EGFR suppressive assay, was the most effective derivative against BRAF^V600E^ (IC_50_ = 92 ± 07 nM). These results suggest that BRAF may not be a viable target for this group of organic molecules.

### 2.3. Docking Study

The most effective molecules **5a**, **5d**, **5e**, **7a**, and **7e** were selected for further study of their probability of interaction modes through active sites of EGFR and BRAF using erlotinib as a reference compound. Molecular docking simulations inside the EGFR active site were used to evaluate the “Scaffold A” group’s potency as EGFR inhibitors, as shown in Table 3. Compound **5a** revealed the greatest docking scores −7.05 and −6.69 (S; kcal/mol) within the five test compounds compared to the reference compound (erlotinib) at −7.06 and −8.02, respectively.

The five test compounds’ best docking positions with the co-crystallized ligand (erlotinib) revealed stability of the compounds within the cavity of the active sites with a number of H-bonds and pi-H hydrophobic interactions with several residues of amino acids around the active site, as illustrated in Figure 4 (See also Appendix A) Compound **5a** within the active sites of EGFR has three hydrogen bonds with Met 742 and Asp 831 whereas erlotinib forms two hydrogen bonds with Met 769 and a water molecule and a pi-H hydrophobic interaction with Lys 721.

On the other hand, compound **5a** within the active sites of BRAF has two hydrogen bonds with Ser 536 and Gly 466 and a pi-H hydrophobic interaction Phe 583. The order of the docking scores fitted with the results of the biochemical tests. Additionally, Substitution of the oxygen atom at position 2 of compounds **5a** and **7a** with a sulphur atom results in compound **5c** and **7c**, respectively where (R = Me, X = S), which showed a reduction in the docking scores (S) in both EGFR active sites (−4.75, −4.95) and BRAF active sites (3.88, 4.92) due to an absence/decrease in hydrogen bonds, Table 4, Figure 5 (See also Appendix A). Therefore, it is obvious that the stated docking results are in agreement with the biological findings.

## 3. Experimental

### 3.1. Chemistry

General details: Refer to Appendix A

Compounds 5,6-diaminouracils **3a**–**f** were prepared according to the reported method [21,26,27].

#### 3.1.1. General Procedures for the Synthesis of 2,7-Dibromo-3′-ethyl-7′,9′-dihydrospiro-[fluorene-9,8′-purines] **5a**–**e**

A mixture of 5,6-diaminouracils (**3a**–**e**) (0.9 mmol) and 2,7-Dibromo-9-fluorenone (**4**) (0.9 mmol) and drops of DMF were heated in fusion for 20 min. The residue was treated with an appropriate amount of ethanol. The precipitate was washed with methanol, filtered, and crystallized from DMF.

2,7-Dibromo-3′-methyl-7′,9′-dihydrospiro[fluorene-9,8′-purine]-2′,6′(1′*H*,3′*H*)-dione (**5a**)

Deep orange solid, Yield: 64%; mp > 300 °C; IR (KBr) ν_max_ (cm^−1^): 3162, 3120 (NH), 3049 (CH Ar), 2839 (CH aliph), 1695 (C=O), 1489 (C=C), 735, 729 (monosubstituted phenyl); ^1^H NMR (400 MHz, DMSO-*d_6_*) δ_H_ 11.00 (s, 1H, NH), 8.02 (s, 1H, Ar), 7.74 (d, *J* = 7.8 Hz, 2H, Ar), 7.70–7.62 (m, 1H, Ar), 7.57–7.56 (m, 2H, Ar), 7.50 (s, 1H, NH), 7.19 (s, 1H, NH), 3.38 (s, 3H, CH_3_) ppm. ^13^C NMR (DMSO-*d_6_*, 100 MHz) δ_C_: 153.7, 153.4, 151.6, 149.3, 140.9, 138.9, 137.7, 134.2, 132.2, 132.0, 130.6, 125.3, 121.8, 121.5, 121.1, 120.7, 102.1, 29.6 ppm. MS: *m*/*z* (rel. int.) = 480 (M^+^ + 4, 22), 478 (M^+^ + 2, 11), 476 (M^+^, 21), 438 (100), 302 (91), 129 (47). Anal. Calcd for C_18_H_12_Br_2_N_4_O_2_ (476.12): C, 45.41; H, 2.54; N, 11.77; Found: C, 45.59; H, 2.70; N, 11.98%.

2,7-Dibromo-3′-ethyl-7′,9′-dihydrospiro[fluorene-9,8′-purine]-2′,6′(1′*H*,3′*H*)-dione (**5b**)

Red solid, Yield: 68%; mp > 300 °C; IR (KBr) ν_max_ (cm^−1^): 3196, 3167 (NH), 3075 (CH Ar), 2977 (CH aliph), 1707, 1622 (C=O), 1516 (C=C), 759, 733 (monosubstituted phenyl); ^1^H NMR (400 MHz, DMSO-*d_6_*) δ_H_ 11.01 (s, 1H, NH), 8.04 (s, 1H, Ar), 7.79 (d, *J* = 7.9, 2H, Ar), 7.60 (d, *J* = 7.9 Hz, 2H, Ar), 7.54 (s, 2H, Ar, NH), 7.18 (s, 1H, NH), 3.99 (q, *J* = 6.9 Hz, 2H, CH_2_), 1.21 (t, *J* = 6.9 Hz, 3H, CH_3_) ppm. ^13^C NMR (DMSO-*d*_6_, 100 MHz) δ_C_ 153.9, 153.2, 152.1, 149.5, 141.3, 139.4, 138.2, 134.7, 132.6, 132.5, 131.1, 125.8, 122.3, 122.0, 121.6, 121.2, 102.4, 37.8, 13.5 ppm. MS: *m*/*z* (rel. int.) = 494 (M^+^ + 4, 44), 492 (M^+^ + 2, 63), 490 (M^+^, 35), 430 (52), 428 (87), 345 (35), 343 (31), 341 (26), 309 (100), 294 (69), 174 (83). Anal. Calcd for C_19_H_14_Br_2_N_4_O_2_ (490.16): C, 46.56; H, 2.88; N, 11.43; Found: C, 46.82; H, 3.07; N, 11.65%.

2,7-Dibromo-3′-methyl-2′-thioxo-2′,3′,7′,9′-tetrahydrospiro[fluorene-9,8′-purin]-6′(1′*H*)-one (**5c**)

Brown solid, Yield: 60%; mp: 298–300 °C; IR (KBr) ν_max_ (cm^−1^): 3184, 3141 (NH), 3050 (CH Ar), 2980, 2837 (CH aliph), 1650 (C=O), 1498 (C=C), 761, 725 (monosubstituted phenyl); ^1^H NMR (400 MHz, DMSO-*d_6_*) δ_H_ 12.42 (s, 1H, NH), 8.04 (s, 1H, Ar), 7.79 (d, *J* = 7.9 Hz, 2H, Ar), 7.67–7.59 (m, 2H, Ar), 7.57 (s, 2H, Ar, NH), 7.22 (s, 1H, NH), 3.86 (s, 3H, CH_3_) ppm. ^13^C NMR (DMSO-*d*_6_, 100 MHz) δ_C_ 173.7, 154.1, 152.4, 150.9, 140.5, 139.3, 138.1, 134.2, 132.9, 132.8, 130.7, 125.8, 122.1, 121.8, 121.3, 120.9, 105.7, 36.5 ppm. MS: *m*/*z* (rel. int.) = 496 (M^+^ + 4, 18), 492 (M^+^, 36), 470 (16), 468 (40), 466 (33), 464 (30), 308 (72), 387 (70), 281 (100), 160 (58). Anal. Calcd for C_18_H_12_Br_2_N_4_OS (492.19): C, 43.93; H, 2.46; N, 11.38; Found: C, 44.17; H, 2.62; N, 11.60%.

3’-Benzyl-2,7-dibromo-7′,9′-dihydrospiro[fluorene-9,8′-purine]-2′,6′(1′*H*,3′*H*)-dione (**5d**)

Yellow solid, Yield: 61%; mp >300 °C; IR (KBr) ν_max_ (cm^−1^): 3154 (NH), 3049, 3030 (CH Ar), 2833 (CH aliph), 1682 (C=O), 1486 (C=C), 756, 735 (monosubstituted phenyl); ^1^H NMR (400 MHz, DMSO-*d_6_*) δ_H_ 11.69 (s, 1H, NH), 8.06–8.03 (m, 1H, Ar), 7.90–7.81 (m, 2H, Ar), 7.63–7.58 (m, 2H, Ar), 7.50–7.46 (m, 3H, Ar, NH), 7.40–7.27 (m, 5H, Ar, NH), 5.36–5.19 (dd, *2*H, CH_2_) ppm. ^13^C NMR (DMSO-*d*_6_, 100 MHz) δ_C_ 154.9, 154.0, 151.9, 151.0, 148.1, 145.6, 139.6, 138.4, 137.6, 135.3, 129.1 (2), 129.0 (2), 128.9, 128.2, 127.4, 126.8, 124.0, 123.5, 122.1 (2), 100.0, 46.4 ppm. MS: *m*/*z* (rel. int.) = 556 (M^+^ + 4, 16), 554 (M^+^ + 2, 25), 552 (M^+^, 10), 323 (28), 321 (48), 319 (37), 317 (29), 270 (94), 223 (100), 162 (42). Anal. Calcd for C_24_H_16_Br_2_N_4_O_2_ (552.23): C, 52.22; H, 2.92; N, 10.15; Found: C, 52.46; H, 3.17; N, 10.32%.

2,7-Dibromo-3′-(2-chlorobenzyl)-7′,9′-dihydrospiro[fluorene-9,8′-purine]-2′,6′(1′*H*,3′*H*)-dione (**5e**)

Bright orange solid, Yield: 59%; mp > 300 °C; IR (KBr) ν_max_ (cm^−1^): 3183, 3156 (NH), 3056, 3033 (CH Ar), 2837 (CH aliph), 1683 (C=O), 1484 (C=C), 756, 730 (monosubstituted phenyl); ^1^H NMR (400 MHz, DMSO-*d_6_*) δ_H_ 11.21 (s, 1H, NH), 8.06 (s, 1H, Ar), 7.84–7.81 (m, 2H, Ar), 7.63 (d, *J* = 6.9 Hz, 2H, Ar), 7.58 (s, 2H, Ar, NH), 7.55–7.53 (m, 1H, Ar), 7.37–7.35 (m, 2H, Ar), 7.31 (s, 1H, NH), 7.10–7.08 (m, 1H, Ar), 5.20 (s, 2H, CH_2_) ppm. ^13^C NMR (DMSO-*d*_6_, 100 MHz) δ_C_: 153.6, 153.3, 151.6, 148.5, 148.1, 144.1, 143.9, 142.6, 142.4, 138.4, 138.0, 134.0, 127.8, 127.7, 124.1, 123.2, 100.0, 46.5 ppm. MS: *m*/*z* (rel. int.) = 590 (M^+^ + 4, 19), 588 (M^+^ + 2, 17), 586 (M^+^, 74). Anal. Calcd for C_24_H_15_Br_2_ClN_4_O_2_ (586.67): C, 49.14; H, 2.58; N, 9.55; Found: C, 49.08; H, 2.69; N, 9.73%.

#### 3.1.2. General Procedures for the Synthesis of Acenaphtho[1,2-*g*]pteridines (**7a**–**f**)

Method A: A mixture of 5,6-diaminouracils **3a**–**f** (1.2 mmol) and acenaphthoquinone (**6**) (1.2 mmol) in acetic acid (3 mL) was heated under reflux for 4 h. The formed precipitate was filtered, washed with ethanol and recrystallized from acetic acid.

Method B: A mixture of 5,6-diaminouracils **3a**–**f** (1.2 mmol) and acenaphthoquinone (**6**) (1.2 mmol) was heated under fusion with drops of DMF for 15 min. An adequate amount of ethanol was added to the residue, the precipitate was filtered and washed with methanol.

8-Methylacenaphtho[1,2-*g*]pteridine-9,11(8*H*,10*H*)-dione (**7a**)

Canary yellow solid, Yield: method A: 65%, method B: 73%; mp > 300 °C [28]; IR (KBr) ν_max_ (cm^−1^): 3170 (NH), 3057 (CH Ar), 2961(CH aliph), 1707, 1674 (C=O), 1449 (C=C); ^1^H NMR (400 MHz, DMSO-*d_6_*) δ_H_ 11.97 (s, 1H, NH), 8.45 (d, *J* = 7.0 Hz, 1H, Ar), 8.38 (d, *J* = 8.2 Hz, 1H, Ar), 8.35 (d, *J* = 7.0 Hz, 1H, Ar), 8.27 (d, *J* = 8.2 Hz, 1H, Ar), 7.94 (m, 2H, Ar), 3.64 (s, 3H, CH_3_) ppm. ^13^C NMR (DMSO-*d*_6_, 100 MHz) δ_C_ 160.8, 150.6, 149.2, 148.0, 133.8, 132.2, 130.8, 130.3, 130.1, 129.8, 129.7, 129.6, 127.6, 125.9, 124.8, 123.1, 29.0 ppm. MS: *m*/*z* (rel. int.) = 302 (M^+^, 11), 270 (64), 212 (45), 162 (100), 65 (67). Anal. Calcd for C_17_H_10_N_4_O_2_ (302.29): C, 67.55; H, 3.33; N, 18.53; Found: C, 67.68; H, 3.49; N, 18.80%.

8-Ethylacenaphtho[1,2-*g*]pteridine-9,11(8*H*,10*H*)-dione (**7b**)

Canary yellow solid, Yield: method A: 64%, method B: 71%; mp > 300 °C; IR (KBr) ν_max_ (cm^−1^): 3174 (NH), 3080 (CH Ar), 2934 (CH aliph), 1689 (C=O), 1499 (C=C); ^1^H NMR (400 MHz, DMSO-*d_6_*) δ_H_ 11.95 (s, 1H, NH), 8.48 (d, *J* = 6.9 Hz, 1H, Ar), 8.39 (d, *J* = 8.2 Hz, 1H, Ar), 8.36 (d, *J* = 6.9 Hz, 1H, Ar), 8.29 (d, *J* = 8.2 Hz, 1H, Ar), 7.99–7.92 (m, 2H, Ar), 4.37 (q, *J* = 7.0 Hz, 2H, CH_2_), 1.33 (t, *J* = 7.0 Hz, 3H, CH_3_) ppm. ^13^C NMR (100 MHz, DMSO) δ_C_ 160.7, 158.0, 155.7, 150.1, 132.2, 130.8, 130.4, 130.0, 129.9, 129.8, 129.7, 129.5, 124.7, 123.0, 25.1, 13.9 ppm. MS: *m*/*z* (rel. int.) = 316 (M+, 21), 308 (49), 299 (73), 252(100), 251 (82), 57 (72), 56 (76). Anal. Calcd for C_18_H_12_N_4_O_2_ (316.32): C, 68.35; H, 3.82; N, 17.71; Found: C, 68.17; H, 3.98; N, 17.98%.

8-Methyl-9-thioxo-9,10-dihydroacenaphtho[1,2-*g*]pteridin-11(8*H*)-one (**7c**)

Yellow solid, Yield: method A: 71%, method B: 79%; mp > 300 °C; IR (KBr) ν_max_ (cm^−1^): 3210 (NH), 3044 (CH Ar), 2935 (CH aliph), 1712 (C=O), 1488 (C=C); ^1^H NMR (400 MHz, DMSO-*d_6_*) δ_H_ 13.09 (s, 1H, NH), 8.46–8.25 (m, 4H, Ar), 7.98–7.92 (m, 2H, Ar), 4.08 (s, 3H, CH_3_) ppm. MS: *m*/*z* (rel. int.) = 318 (M^+^, 49), 315 (46), 295 (100), 277 (95), 133 (99), 83 (67). Anal. Calcd for C_17_H_10_N_4_OS (318.35): C, 64.14; H, 3.17; N, 17.60; Found: C, 64.31; H, 3.40; N, 17.86%.

8-Benzylacenaphtho[1,2-*g*]pteridine-9,11(8*H*,10*H*)-dione (**7d**)

Deep orange solid, Yield: method A: 70%, method B: 78%; mp > 300 °C [28]; IR (KBr) ν_max_ (cm^−1^): 3168 (NH), 3044 (CH Ar), 2918 (CH aliph), 1696, 1673 (C=O), 1499 (C=C); ^1^H NMR (400 MHz, DMSO-*d_6_*) δ_H_ 12.08 (s, 1H, NH), 8.35 (d, *J* = 7.2 Hz, 2H, Ar), 8.32–8.28 (m, 1H, Ar), 8.23 (d, *J* = 8.6 Hz, 1H, Ar), 7.92–7.89 (m, 2H, Ar), 7.53 (d, *J* = 8.6 Hz, 2H, Ar), 7.33 (t, *J* = 7.8 Hz, 2H, Ar), 7.25 (t, *J* = 7.8 Hz, 1H, Ar), 5.48 (s, 2H, CH_2_) ppm. ^13^C NMR (DMSO-*d*_6_, 100 MHz) δ_C_ 160.7, 155.5, 150.6, 148.4, 137.7, 133.8, 132.3, 130.7, 130.2, 130.1, 129.8, 129.7, 129.5, 128.9, 128.4, 127.7, 125.9, 124.8, 123.1, 44.8 ppm. MS: *m*/*z* (rel. int.) = 378 (M^+^, 35), 313 (68), 210 (53), 205 (97), 193 (100), 115 (60). Anal. Calcd for C_23_H_14_N_4_O_2_ (378.39): C, 73.01; H, 3.73; N, 14.81; Found: C, 72.93; H, 3.94; N, 14.98%.

8-(2-Chlorobenzyl)acenaphtho[1,2-*g*]pteridine-9,11(8*H*,10*H*)-dione (**7e**)

Light yellow solid, Yield: method A: 60%, method B: 69%; mp > 300 °C; IR (KBr) ν_max_ (cm^−1^): 3168 (NH), 3045 (CH Ar), 2845 (CH aliph), 1714, 1679 (C=O), 1497 (C=C); ^1^H NMR (400 MHz, DMSO-*d_6_*) δ_H_ 12.16 (s, 1H), 8.39 (d, *J* = 7.6 Hz, 1H, Ar), 8.36 (d, *J* = 8.8 Hz, 1H, Ar), 8.31 (d, *J* = 7.6 Hz, 1H, Ar), 8.29 (d, *J* = 8.8 Hz, 1H, Ar), 7.96–7.90 (m, 2H, Ar), 7.55 (d, *J* = 7.4 Hz, 1H, Ar), 7.35–7.28 (m, 2H, Ar), 7.22 (t, *J* = 7.4 Hz, 1H, Ar), 5.57 (s, 2H, CH_2_) ppm. ^13^C NMR (DMSO-*d*_6_, 100 MHz) δ_C_ 158.3, 151.3, 148.6, 146.2, 141.9, 134.7, 132.2, 130.7, 130.3, 130.2, 130.1, 129.7, 129.6, 129.2, 128.7, 128.6, 128.2, 127.9, 127.8, 126.3, 46.5 ppm. MS: *m*/*z* (rel. int.) = 414 (M + 2, 12), 412 (M+, 49), 334 (16), 332 (73), 185 (77), 84 (38), 82 (100). Anal. Calcd for C*_23_*H_13_ClN_4_O_2_ (412.83): C, 66.92; H, 3.17; N, 13.57; Found: C, 66.80; H, 3.41; N, 13.79%.

Acenaphtho [1,2-*g*]pteridine-9,11(8*H*,10*H*)-dione (**7f**)

Yellow solid, Yield: method A: 69%, method B: 77%; mp > 300 °C; IR (KBr) ν_max_ (cm^−1^): 3208 (NH), 3036 (CH Ar), 2819 (CH aliph), 1688, 1643 (C=O), 1486 (C=C); ^1^H NMR (400 MHz, DMSO-*d_6_*) δ_H_ 11.63 (s, 1H, NH), 11.29 (s, 1H, NH), 8.34–8.28 (m, 1H, Ar), 8.17 (m, 1H, Ar), 8.09 (m, 2H, Ar), 7.93–7.76 (m, 2H, Ar) ppm. ^13^C NMR (DMSO-*d*_6_, 100 MHz) δ 161.8, 156.4, 150.3, 149.1, 133.8, 132.8, 132.1, 131.0, 130.3, 129.8, 129.7, 129.5, 129.0, 124.5, 122.7, 121.7 ppm. MS: *m*/*z* (rel. int.) = 288 (M^+^, 17), 264 (53), 186 (50), 107 (62), 63 (100). Anal. Calcd for C_16_H_8_N_4_O_2_ (288.27): C, 66.67; H, 2.80; N, 19.44; Found: C, 66.73; H, 2.91; N, 11.65%.

### 3.2. Biology

#### 3.2.1. Cell Viability Assay

The normal human mammary gland epithelial (MCF-10A) cell line was used to test the viability of new compounds [29,30]. See Appendix A.

#### 3.2.2. Anti-Proliferative Assay

The antiproliferative activity of compounds **5a**–**e** and **7a**–**f** was tested against the four human cancer cell lines Panc-1 (pancreatic cancer cell line), MCF-7 (breast cancer cell line), HT-29 (colon cancer cell line), and A-549 (lung cancer cell line) using the MTT assay and erlotinib as the reference drug [11,12,13,14,15,16,17,18,19,20,21,22,23,24,25,26,27,28,29,30,31]. See Appendix A.

#### 3.2.3. EGFR Inhibitory Assay

Compounds **5a, 5d, 5e 7a**, and **7e** were tested for EGFR inhibitory activity as a potential target for their antiproliferative activity [34,35]. See Appendix A.

#### 3.2.4. BRAF^V600E^ Inhibitory Assay

Compounds **5a**, **5d**, **5e 7a**, and **7e** were further tested for BRAF^V600E^ inhibitory activity as a potential target for their antiproliferative activity [36]. See Appendix A.

### 3.3. Protocol of Docking Studies

The automated docking simulation study was performed using Molecular Operating Environment (MOE^®^) version 2014.09. The X-ray crystallographic structure of the target EGFR and BRAF was obtained from the protein data bank (PDB: 1M17, 5JRQ), respectively. The target compounds were constructed in a three-dimensional model using the builder interface of the MOE^®^ program. After checking their structures and the formal charges on atoms by two-dimensional depiction, the following steps were carried out: The target compounds were subjected to a conformational search. All conformers were subjected to energy minimization; all the minimizations were performed with MOE until a RMSD gradient of 0.01 Kcal/mole and RMS distance of 0.1 Å with MMFF94X force-field and the partial charges were automatically calculated. The protein was prepared for docking studies by adding hydrogen atoms to the system with their standard geometry. The atoms connection and type were checked for any errors with automatic correction. Selection of the receptor and its atoms potential were fixed. MOE Alpha Site Finder was used for the active site search in the enzyme structure using all default items. Dummy atoms were created from the obtained alpha spheres [37,38].

## 4. Conclusions

In summary, two sets of purine/pteridine-based analogues **5a**–**e** and **7a**–**f** were designed and synthesised. The newly synthesised compounds were tested for anti-proliferative activity, and compounds **5a**, **5e**, and **7e** were found to be the most effective. SAR analysis revealed that replacing the oxygen atom in position 2 of compounds **5a** or **7a** with a sulphur atom resulted in compounds **5c** and **7c**, which had significantly lower anti-proliferative activity. When compared to erlotinib’s IC_50_ value of 80 nM, compounds **5a**, **5e**, and **7e** demonstrated promising EGFR inhibitory activity, with IC_50_ values of 87 nM, 98 nM, and 92 nM, respectively. These findings suggest that EGFR could be a potential target for compounds **5a**, **5e**, and **7e**, which would necessitate more in-depth structural investigation to identify a lead compound for future development. Furthermore, the molecular docking study was performed on the EGFR and BRAF^V600E^ active sites revealing good interactions with the enzymes.

## Data Availability

The data will be provided upon request.

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
