# Peer review of "Design, Synthesis, and Anti-Proliferative Action of Purine/Pteridine-Based Derivatives as Dual Inhibitors of EGFR and BRAFV600E"

_pharmaceuticals, 2023, doi:10.3390/ph16050716_

Round 1
Reviewer 1 Report
- The design of the molecules is highly speculative but some compounds are bioactive so...why not????
- The chemical synthesis is OK but the quality of the NMR spectra in the SI is very poor and must be improved with higher peaks to be able to see them. Actually it's impossible to read such spectra, especially the 13C spectra see for instance 5d, 5e, 7b, 7e and 7c (even no spectrum!)
- In the same way, the docking poses are impossible to read both in text and SI. Good quality Fig 1 and Fig2 must be provided.
- Finally, for the biological studies, these must be analyzed by a biologist expert in these areas. It is not possible for me as a chemist.
Minor revisions required
Author Response
Comments and Suggestions for Authors
- Reviewer 1:
- The design of the molecules is highly speculative but some compounds are bioactive so...why not????
Response: The design of this work based on our published paper in Bioorganic Chemistry,116, 2021, 105344 using Quinolone scaffold, herein we plan to use purine-based and pteridine-based scaffolds and screened their activities.
- The chemical synthesis is OK but the quality of the NMR spectra in the SI is very poor and must be improved with higher peaks to be able to see them. Actually it's impossible to read such spectra, especially the 13C spectra see for instance 5d, 5e, 7b, 7e and 7c (even no spectrum!)
Response: Unfortunately, the mentioned compounds dissolve hardly in DMSO, the peaks are not strong enough even after running the NMR machine for a long time, tried different NMR instruments and use different NMR solvents. The charts of listed compounds were reinserted in higher quality.
In the same way, the docking poses are impossible to read both in text and SI. Good quality Fig 1 and Fig2 must be provided.
Response: Thanks for this comment; we improved quality of the listed figures.
Finally, for the biological studies, these must be analyzed by a biologist expert in these areas. It is not possible for me as a chemist.
- Comments on the Quality of English Language
We polish the language of the manuscript.
Minor revisions required
Response: English of manuscript is refined
Submission Date
31 March 2023
Date of this review
03 Apr 2023 14:39:53
Reviewer 2 Report
In this submitted manuscript " Design, Synthesis, and Antiproliferative Action of Purine/pteridine-based Derivatives as Dual inhibitors of EGFR and BRAFV600E” the authors have successfully synthesized a series of new purine/pteridine analogues to increase the antibroliferative reactivity. From an organic point of view, the scope is exceedingly limited. In addition, the results of anticancer activity and inhibition of EGFR and BRAFV600E are not highlighted in comparison to the standard drug (Erlotinib). I consider that mechanism of action studies might be included. I thus recommend its publication after the following points have been dealt with:
1)The manuscript must be reconstructed according the templet of Pharmaceuticals such as title, the first litters must be written in capital, the order of sub title in results and discussion part must be the same order of title and sub title in experimental part. Also, all title and sub title must have number, the references should be writing according to the template and so on.
2)The numbers of the figures and schemes must be revised in the whole document
3)In the results and discussion part, in the test we have 1a-f and 2a-f and in the scheme you have 1a-e …5a-e ???
4)Scheme 2 should be add after compounds 5a-e in reasonable yields (59-68%) in chemistry part.
5)Since the activity is linked to the alkyl part and 2-Cl-Bn seems to be the most active derivative I’m wondering if the authors have test others derivative of benzyl with hydrogen bond acceptor groups such as 2-methoxy-Bn if not I’m requesting to add this experience
According to the above comments, the article needs major revision. So I accept the article to be publishing in Pharmaceuticals after correction
Author Response
The investigation of novel EGFR and BRAFV600E dual inhibitors tended to serve as targeted cancer treatment. Two sets of purine/pteridine-based derivatives were designed and synthesized as EGFR/BRAFV600E dual inhibitors. The majority of the compounds demonstrated promising antiproliferative activity on the cancer cell lines tested. Compounds 5a, 5e, and 7e of purine-based and pteridine-based scaffolds were identified as the most potent hits in antiproliferative screening, with GI50 values of 38 nM, 46 nM, and 44 nM, respectively. 5a, 5e, and 7e demonstrated promising EGFR inhibitory activity, with IC50 values of 87 nM, 98 nM, and 92 nM, respectively, when compared to erlotinib's IC50 value of 80 nM. According to the results of the BRAFV600E inhibitory assay, BRAFV600E may not be a viable target for this class of organic compounds. Finally, molecular docking studies were carried out at the EGFR and BRAFV600E active site to suggest possible binding modes
Bottom of Form
Top of Form
Comments and Suggestions for Authors
In this submitted manuscript " Design, Synthesis, and Antiproliferative Action of Purine/pteridine-based Derivatives as Dual inhibitors of EGFR and BRAFV600E” the authors have successfully synthesized a series of new purine/pteridine analogues to increase the antibroliferative reactivity. From an organic point of view, the scope is exceedingly limited. In addition, the results of anticancer activity and inhibition of EGFR and BRAFV600E are not highlighted in comparison to the standard drug (Erlotinib). I consider that mechanism of action studies might be included. I thus recommend its publication after the following points have been dealt with:
- The manuscript must be reconstructed according the templet of Pharmaceuticals such as title, the first litters must be written in capital, the order of sub title in results and discussion part must be the same order of title and sub title in experimental part. Also, all title and sub title must have number, the references should be writing according to the template and so on.
Response: Done
2) The numbers of the figures and schemes must be revised in the whole document
Response: Thank you for the comment, adjusted as advised.
3) In the results and discussion part, in the test we have 1a-f and 2a-f and in the scheme you have 1a-e …5a-e ???
Response: Thank you for the valuable comment, adjusted as advised.
4) Scheme 2 should be add after compounds 5a-e in reasonable yields (59-68%) in chemistry part.
Response: Thank you for the comment, the yields for compounds 5a-e were added to scheme 1 as advised.
5) Since the activity is linked to the alkyl part and 2-Cl-Bn seems to be the most active derivative I’m wondering if the authors have test others derivative of benzyl with hydrogen bond acceptor groups such as 2-methoxy-Bn if not I’m requesting to add this experience
Response: your valuable note would be taken in consideration for further work in future.
According to the above comments, the article needs major revision. So I accept the article to be publishing in Pharmaceuticals after correction.
Submission Date
31 March 2023
Date of this review
19 Apr 2023 17:49:00
Reviewer 3 Report
· In the present study, the authors described the synthesis of some novel substituted purine/pteridine-based derivatives and investigated for antiproliferative activities as EGFR/BRAFV600E dual inhibitors. The experiments and writing are both convincing. Although the statistics significantly support the conclusions and the references section is relevant, the research writing needs to be thoroughly examined and should be reviewed carefully and all of the figures need to be of higher quality.
· I suggested that the introduction be updated to incorporate more details about the study of cancer molecular docking.
· In the title: the word synthesis begins with a lowercase letter.
· In page 1: ”AbstractThe” should be corrected to [Abstract The].
· In page 3: ”Fig. 1” should be corrected to [Figure 1].
· In page 4: ”Figure 6” should be corrected to [Figure 2] and ”Fig. III” should be corrected to [Figure 3].
· In page 4: The structures of 5a-e (Scaffold A), it lacks )R) should be corrected to (N-R) in figure 3.
· In page 5: It must be mentioned that the intermediates No. VII and VIII follow a scheme 2.
· In page 6: acenaphthoquinone 6 should be bold, and corrected to 6.
· In page 7: ” Scheme 2” should be corrected to [Scheme 3].
· In page 8: The word "Scaffold A" is in brackets. Also, “scaffold B” in page 9.
· In page 12: Figure 1. 2D and 3D Interaction diagram of 5a, 7e and Erlotinib within EGFR (PDB ID:1M17) should be corrected to [Figure 4]. Also, the 2D and 3D Interaction diagrams needs to be more clearer.
· In page 13: Figure 2. 2D and 3D Interaction diagram of 5a, 7e and Erlotinib within BRAFV600E (PDB ID: 5JRQ) should be corrected to [Figure 5]. The interaction diagrams also require improvement in clarity.
· In the experimental section, compounds names with a "H" should be italicized. For ex. “Dibromo-3'-methyl-7',9'-dihydrospiro[fluorene-9,8'-purine]-2',6'(1'H,3'H)dione” should be corrected to [Dibromo-3'-methyl-7',9'-dihydrospiro[fluorene-9,8'-purine]-2',6'(1'H,3'H)dione] and the same goes for the remaining compounds title.
· To better explain their findings and the improvement over the reference medications they utilized for the pharmacological testing, the authors could elaborate on the data presented in the anti-proliferative tables and the significance of all that data in the biological evaluation.
· Finally, IR spectra must be included for additional validation.
English language needs only little editing
Author Response
- Reviewer 3:
- In the present study, the authors described the synthesis of some novel substituted purine/pteridine-based derivatives and investigated for antiproliferative activities as EGFR/BRAFV600Edual inhibitors. The experiments and writing are both convincing. Although the statistics significantly support the conclusions and the references section is relevant,
Response: The authors thank Reviewer #2 for this positive feedback.
- The research writing needs to be thoroughly examined and should be reviewed carefully and all of the figures need to be of higher quality.
Response: Done
- I suggested that the introduction be updated to incorporate more details about the study of cancer molecular docking.
Response: introduction is updated as required.
- In the title: the word synthesis begins with a lowercase letter.
Response: Adjusted as advised.
- In page 1: ”AbstractThe” should be corrected to [Abstract The].
Response: Adjusted as advised.
- In page 3: ”Fig. 1” should be corrected to [Figure 1].
Response: Thank you for the comment, it was done as advised.
- In page 4: ”Figure 6” should be corrected to [Figure 2] and ”Fig. III” should be corrected to [Figure 3].
Response: Thank you very much for the comment, it was done as advised.
- In page 4: The structures of 5a-e(Scaffold A), it lacks )R) should be corrected to (N-R) in figure 3.
Response: Thank you very much for the comment, modified as advised.
- In page 5: It must be mentioned that the intermediates VII and VIII follow a scheme 2.
Response: Thank you for the suggestion, scheme 2 was added as advised.
- In page 6: acenaphthoquinone 6 should be bold, and corrected to 6.
Response: Adjusted as advised.
- In page 7: ” Scheme 2” should be corrected to [Scheme 3].
Response: Modified as advised.
- In page 8: The word "Scaffold A"is in brackets. Also, “scaffold B” in page
Response: Thank you for the comment, adjusted as advised.
- In page 12: Figure 1. 2D and 3D Interaction diagram of 5a, 7e and Erlotinibwithin EGFR (PDB ID:1M17) should be corrected to [Figure 4]. Also, the 2D and 3D Interaction diagrams needs to be more clearer.
Response: Thank you for the comment, adjusted as advised.
- In page 13: Figure 2.2D and 3D Interaction diagram of 5a, 7e and Erlotinib within BRAFV600E (PDB ID: 5JRQ) should be corrected to [Figure 5]. The interaction diagrams also require improvement in clarity.
Response: Thank you for the comment, adjusted as advised.
- In the experimental section, compounds names with a "H" should be italicized. For ex. “Dibromo-3'-methyl-7',9'-dihydrospiro[fluorene-9,8'-purine]-2',6'(1'H,3'H)dione” should be corrected to [Dibromo-3'-methyl-7',9'-dihydrospiro[fluorene-9,8'-purine]-2',6'(1'H,3'H)dione] and the same goes for the remaining compounds title.
Response: Thank you for the comment, adjusted as advised.
- To better explain their findings and the improvement over the reference medications they utilized for the pharmacological testing, the authors could elaborate on the data presented in the anti-proliferative tables and the significance of all that data in the biological evaluation.
Response: Actually, we used the obtained biological results to write on the structure activity relationship as mentioned in the discussion section
- Finally, IR spectra must be included for additional validation.
Response: Thank you for the suggestion, IR was included as advised.
- Comments on the Quality of English Language
English language needs only little editing
Response: English of manuscript is refined
Submission Date
31 March 2023
Date of this review
11 Apr 2023 16:24:52
Round 2
Reviewer 1 Report
Not much improvment in the quality of NMR spectra. If not soluble enough for good NMR is there not any problem for the quality of the biological studies.
For the docking studies there is no solubility problem and the quality must be improved, it's just a matter of computer system.
no further comment
Reviewer 2 Report
I accept in present form